# Anti-Citrullinated Peptide Antibodies Control Oral *Porphyromonas* and *Aggregatibacter species* in Patients with Rheumatoid Arthritis

**DOI:** 10.3390/ijms232012599

**Published:** 2022-10-20

**Authors:** Marina I. Arleevskaya, Eugenia A. Boulygina, Regina Larionova, Shamil Validov, Olga Kravtsova, Elena I. Shagimardanova, Lourdes Velo, Geneviève Hery-Arnaud, Caroline Carlé, Yves Renaudineau

**Affiliations:** 1Central Research Laboratory, Kazan State Medical Academy, 420012 Kazan, Russia; 2Institute of Fundamental Medicine and Biology, Kazan (Volga Region) Federal University, 420012 Kazan, Russia; 3Department of Bacteriology, Brest University Hospital, UBO, INSERM UMR 1078, 29200 Brest, France; 4Laboratory of Immunology, CHU Purpan Toulouse, INSERM U1291, CNRS U5051, University Toulouse III, 31062 Toulouse, France

**Keywords:** rheumatoid arthritis, ACPA, clinical suspect arthralgia, first degree relatives, oral microbiome, *Porphyromonas* sp., *Aggregatibacter* sp.

## Abstract

Oral microbiome changes take place at the initiation of rheumatoid arthritis (RA); however, questions remain regarding the oral microbiome at pre-RA stages in individuals with clinically suspect arthralgia (CSA). Two cross-sectional cohorts were selected including 84 Tatarstan women (15 early-RA as compared to individuals with CSA ranging from CSA = 0 [*n* = 22], CSA = 1 [*n* = 19], CSA = 2 [*n* = 11], and CSA ≥ 3 [*n* = 17]) and 42 women with established RA (median: 5 years from diagnosis [IQ: 2–11]). Amplicon sequence variants (ASVs) obtained from oral samples (16S rRNA) were analyzed for alpha and beta diversity along with the abundance at the genus level. A decrease in oral *Porphyromonas* sp. is observed in ACPA-positive individuals, and this predominates in early-RA patients as compared to non-RA individuals irrespective of their CSA score. In the RA-established cohort, *Porphyromonas* sp. and *Aggregatibacter* sp. reductions were associated with elevated ACPA levels. In contrast, no associations were reported when considering individual, genetic and clinical RA-associated factors. Oral microbiome changes related to the genera implicated in post-translational citrullination (*Porphyromonas* sp. and *Aggregatibacter* sp.) characterized RA patients with elevated ACPA levels, which supports that the role of ACPA in controlling the oral microbiome needs further evaluation.

## 1. Introduction

Rheumatoid arthritis (RA) represents a chronic and inflammatory autoimmune disease starting in the mucosa (e.g., oral, digestive, lung), and the microbiome is highly suspected of initiating an immune response leading to RA in genetically predisposed individuals [1,2]. Indeed, oral bacterial analysis has focused on bacterial genera or species present in periodontal disease effective in inducing post-translational citrullination, which subsequently generates anti-citrullinated protein/peptide antibodies (ACPA) that drive autoimmunity and, in turn, RA. One example is *Porphyromonas gingivalis* (Pg), a Gram-negative anaerobe that contains a bacterial peptidyl arginine deiminase (PPAD) capable of citrullinating arginine residues in host proteins [3]. Another example is *Aggregatibacter actinomycetemcomitans* (Aa), a facultative Gram-negative and oral anaerobe, which is also able to trigger, through leukotoxin A release, mammalian peptidyl arginine deiminase (PADI) overexpression from host neutrophils and, in turn, citrullination [4]. However, demonstration of a specific role for oral Pg and Aa in triggering periodontitis and, in turn, RA, relies predominantly on serological approaches [5,6], while results from microbiome analyses are more contrasted. As reviewed in Table 1, some but not all authors have reported a quantitative (alpha diversity) and qualitative (beta diversity) evolution in pre-RA and/or RA individuals when exploring the oral microbiome. Altogether, this supports the concept that the oral microbiome changes, including periodontal bacteria, may be present at the initiation of the disease in pre-RA individuals with clinically suspect arthralgia (CSA), which is the initial aim of the study [7].

Oral microorganism colonization results from multiple parameters, including diet, contact with other individuals/animals, dentition, and hygiene habits, among other factors [19]. In adulthood, ecologic stability is obtained with interspecies collaborations and antagonisms in order to maintain homeostasis [20]. As reported from 200 healthy individuals and 9 oral sites, the human oral microbiome (HOMD; http://www.homd.org (accessed on 12 October 2022)) is complex, with over 700 prokaryotic species. The higher proportion of bacteria is reported to be from the phyla *Firmicutes* (e.g., *Streptococcus* sp., *Gemella* sp.), while smaller proportions are observed in *Bacteroides* (e.g., *Porphyromonas* sp.), *Proteobacteria* (e.g., *Haemophilus* sp., *Serratia* sp.), and *Actinobacteria* (e.g., *Rothia* sp.) [21]. Changes in the oral cavity microbiome have been associated with multiple factors including oral bacterial diseases such as periodontitis, prosthetic implant, tobacco smoking, salivary gland dysfunction, and systemic diseases [22,23].

Accordingly, women from the Tatarstan cohort were selected and non-RA individuals were subdivided according to their CSA score. For this, first-degree relatives (FDR) from RA probands were selected as they responded to one or more criteria of CSA and are at high risk of developing RA: 9.1 cases/1000/year among the Tatarstan cohort [24]. The initial analysis was conducted to compare individuals classified according to their CSA score and compare with early RA. Next, and as modifications were reported not at the CSA stage but at RA onset, results were further confirmed in patients with established RA. Amplicon sequence variants (ASVs) obtained from oral samples (16S rRNA) were analyzed for alpha and beta diversity along with the abundance at the genus level according to the individual group, clinico-biological status, and genetic and individual factors, to highlight associations with oral dysbiosis.

## 2. Results

### 2.1. Characteristics of the Two Cohorts Studied

As reported in Table 2, two cohorts were considered. First, 69 Tatarstan women ranging from CSA = 0 [*n* = 22], CSA = 1 [*n* = 19], CSA = 2 [*n* = 11], and CSA ≥ 3 [*n* = 17] were included and compared with 15 women who had received the diagnosis of RA in the last year. Second, 42 RA women more distant from RA diagnosis (median: 5 years [IQ: 2–11]) were further studied. The majority of the patients with early RA and with established RA presented a joint-28 disease activity score (DAS28) of moderate/high (53.3% and 76.2%, respectively) and elevated levels of ACPA (66.6% and 57.1%), and they were predominantly under methotrexate (MTX) treatment (46.7% and 76.2%); for some of them, this was in combination with glucocorticoids (GC, 6.7% and 33.3%) or biologics (0% and 7.0%).

### 2.2. Porphyromonas sp. Reduction Characterizes Early RA Patients

In order to explore the influence of the oral microbiome at the pre-clinical stage that precedes RA development, buccal swabs were obtained from 69 non-RA individuals subdivided according to their CSA score and 15 early-RA patients. Next, and following 16S rRNA sequence preparation (demultiplexing, adapter removing, and cleaning), sequences were annotated using dada2, and the corresponding SILVA taxonomy file was analyzed at the genus level. The capacity to influence the oral microbiome richness and evenness/consistency (alpha diversity) and microbiota community structure (beta-diversity) was first tested (Figure 1A,B). No differences were observed when comparing by ANOVA Chao1 index for richness (*p* = 0.363) and Shannon’s index for diversity (*p* = 0.615), and when performing Permutational Multivariate Analysis of Variance (PERMANOVA) to test beta diversity between the five groups (*p* = 0.615).

The top 10 genera with highest relative abundance are presented in Figure 1C, which are *Streptococcus* sp. (34.9–45.5%), *Haemophilus* sp. (8.6–10.9%), *Gemella* sp. (3.3–6.9%), *Rothia* sp. (2.7–8.2%), *Serratia* sp. (2.9–6.5%), *Neisseria* sp. (2.1–6.0%), *Granulicatella* sp. (1.0–4.3%), *Prevotella* sp. (2.1–3.8%), *Porphyromonas* sp. (0.2–2.9%), and *Veillonella* sp. (2.5–6.2%). Next, the linear discriminant analysis (LDA) effect size (LEfSe) was used to further explore oral microbiome changes, and one genus reached significance: *Porphyromonas* sp. (*p* = 0.007, LDA = 2.59), which was significantly less abundant in patients with early RA as compared to the four CSA subgroups from non-RA individuals (Figure 1D). When the analysis was performed within *Porphyromonas* sp. at the species and the ASV levels (*n* = 67), no direct correlation was observed including among Pg (five ASVs) and unclassified ASVs when analyzed individually (58 ASVs). Taken together, these data suggest that *Porphyromonas* sp. mouth colonization is affected at the onset of RA as compared to the pre-clinical/CSA phase.

### 2.3. Individual, Genetic and Bio-Clinical Factors

As oral *Porphyromonas* sp. variations may result from socio-demographic factors and individual behavior factors [25], the influence of 12 RA-associated individual factors (childbirth, menopause, familial RA history, low education level, overweight/obesity, active/passive tobacco smoking, and alcohol/fish/coffee consumption) on *Porphyromonas* sp. was tested within the 69 non-RA individuals plus or minus the 15 early-RA patients. As shown in Figure 2A, which presents the -Log10 of the LEfSe q value, *Porphyromonas* sp. abundance was independent from the 12 RA-associated individual factors tested when considering or not early-RA patients.

Next, the same analysis was repeated by testing the genetics (HLA-DRB1 shared epitope [SE]), biological (ACPA, rheumatoid factor [RF], and the erythrocyte sedimentation rate [ESR]), pre-RA/RA symptoms (tender arthralgia, joint arthralgia, morning stiffness ≥ 60 min), disease scores (DAS28-ESR), and MTX use. When considering the 69 non-RA individuals (Figure 2B, white boxes), none of the parameters tested were associated with oral *Porphyromonas* sp. Next, the addition of the 15 early RA to the 69 non-RA (Figure 2B–E) retrieved three negative associations with oral *Porphyromonas* sp.: DAS28-ESR (*p* = 0.01, LDA = −2.3), MTX use (*p* = 0.01, LDA = −2.4), and ACPA positivity (*p* = 0.015, LDA = −2.4).

### 2.4. RA Population

Since the three factors retrieved, which influence *Porphyromonas* sp., predominate among the 15 early-RA patients, we further selected 42 RA women more distant from their RA diagnosis (median: 5 years [IQ: 2–11]) in order to confirm the influence of RA factors on the oral microbiome. As presented in Figure 3A, an oral *Porphyromonas* sp. reduction (q = 0.009, LDA = 2.1) was associated in RA-advanced patients with ACPA detection at elevated levels (≥3 upper limit of normal [ULN]), which was not the case for the other factors tested, including a moderate/high disease activity and the use of MTX.

Finally, and to further support our hypothesis that elevated ACPA levels were associated with oral microbiome changes in RA patients, the RA-advanced patients were subdivided based on an elevated ACPA status. Using heatmap and LEfSe approaches (Figure 3B,D), elevated levels of ACPA were associated with two genera: *Porphyromonas* sp. (*p* = 0.001, LDA = 2.1) and *Aggregatibacter* sp. (*p* = 0.025, LDA = 1.2). Altogether, this supported the proposition that RA-associated oral bacterial genera implicated in post-translational citrullination (*Porphyromonas* sp. and *Aggregatibacter* sp.) are controlled by ACPA when present at elevated levels in RA patients.

## 3. Discussion

Recent epidemiological and serological studies have implicated oral microorganisms, such as Pg and Aa, present in periodontal disease, as possible triggering factors for RA based on their capacity to induce host citrullination. Accordingly, one may expect that an oral dysbiosis may help to determine individuals at high risk of developing RA, such as CSA-positive individuals, which was tested in our study but not confirmed. On the contrary, both *Porphyromonas* sp. and *Aggregatibacter* sp. were affected in RA patients and this was associated with the presence of the serological RA biomarker, ACPA. Such an association seems to be specific as we failed to report an association with other RA-associated risk factors (e.g., HLA-DRB1 SE, tobacco smoking, education level), RA-associated biological factors (e.g., ESR, RF), and CSA/RA clinical presentations (e.g., joint arthralgias, swollen joints, morning stiffness).

Antibodies against lipopolysaccharide (LPS) extracted from Pg (anti-Pg) can be present several years before the clinical onset of RA [6], and their titers are higher when associated with ACPA in FDR from RA probands and of RA patients [26,27]. The association between the Ab response against arginine-specific gingipain, a Pg virulence factor, and the detection of ACPA, tobacco smoking and the presence of the HLA SE was also confirmed in 1974 RA patients [28]. The explanation was provided by the fine analysis of germline monoclonal (m)Ab derived from ACPA-positive plasmocytes, as they are effective in targeting uncitrullinated outer bacterial membrane antigens (e.g., LPS, gingipains) from *Porphyromonas* sp. [29,30]. In other words, and through mAb mutations, germline mAb directed against *Porphyromonas* sp. has the capacity to recognize citrullinated proteins. This cross-reactivity is further promoted by Pg as these bacteria express the PPAD enzyme at its cell wall, allowing the recognition of both membrane and autocitrullinated bacterial proteins [31]. As a consequence, the humoral response against *Porphyromonas* sp. is important in controlling periodontal bacteria, oral inflammation, bacterial dissemination, and inflammatory arthritis, but with the risk of inducing a host immune response to citrullination that will initiate RA development. Regarding Aa, the presence of Ab directed against the LPS of Aa, serotype B, or against its virulence factor, leukotoxin B, have been further reported in RA [5]. The release of leukotoxin A is effective in inducing neutrophil extracellular traps (NETs) that possess plenty of citrullinated histones and DNA such that the NETs are able to engulf the bacteria in order to prevent Aa dissemination [32,33,34]. As a consequence, ACPA capacity to recognize bacterial membrane antigens provides an explanation to our report that ACPA detection is associated with a reduction of *Porphyromonas* sp. and *Aggregatibacter* sp.

An unsolved question is related to the importance of the citrullination process for periodontal bacteria such as *Porphyromonas* sp. and *Aggregatibacter* sp. to drive arthritis and RA. Indeed, this question was addressed in several animal models with conflicting results. In the K/BxN serum arthritis model, Munoz-Atienza et al. reported that the Pg capacity to exacerbate arthritis and to mediate intestinal barrier breakdown is conserved when PPAD is rendered nonfunctional by mutation, which supports independence from citrullination [30]. Similarly, the use of uncitrullinated alpha-enolase from Pg to immunize mice harboring the human HLA DRB1-DR4 SE was reported to be effective in inducing a humoral response, important joint changes, synovial hyperplasia, and erosion that mimic RA [35]. However, using a similar model as reported by Munoz-Atienza et al., Haag et al. have reproduced the humoral response but not arthritogenicity [36]. This is in line with the collagen-induced arthritis murine model in which the Pg capacity to colonize the periodontium and to exacerbate arthritis due to its dissemination in the joints is restricted to PPAD expression [37,38,39].

Compared to culture-based methods that are hampered by species that are difficult to cultivate, the oral 16S RNA microbiome next-generation sequencing analysis approach provides a deeper picture of the bacterial complexity. However, and as reviewed in Table 1, results from oral microbiome analysis are heterogeneous and several parameters need to be taken into consideration when comparing studies: (i) sampling strategies (e.g., oral mucosa, saliva, and subgingival plaque); (ii) amplification process (e.g., 16S RNA versus whole genus shotgun that influences the resolution level [40]; as an example, 16S RNA amplicon sequencing allows good resolution at the genus level, while species and sub-species identification is difficult, as reported for *Porphyromonas* sp. (58/67 ASVs were unclassified) and *Aggregatibacter* sp. (13/19 ASVs were unclassified) in our study [41]); (iii) bioinformatic pipelines (e.g., ASVs are progressively replacing operational taxonomic units [OTU]) [42]; and (iv) population characteristics (e.g., early/chronic RA, ACPA status, therapy). Accordingly, alpha and beta diversity, independent from the CSA and RA status, reported in our study may be explained by our choice to use oral mucosa samples. When reported, alpha and beta diversity differences were described from saliva and subgingival plaque oral microbiomes [18], which concentrate periodontal bacteria. Another important criterion when studying the oral microbiome in pre-RA individuals is their definition, as we failed to associate oral dysbiosis with the clinical parameters that define CSA, as compared to the presence of ACPA when used to define pre-RA. We further reported that *Porphyromonas* sp. and *Aggregatibacter* sp. are less abundant in RA patients as compared to non-RA individuals. This reduction was reported in 4/12 studies, including one using oral mucosa (see Table 1). Furthermore, we failed to report an association for this reduction with (i) RF positivity; (ii) HLA DRB1 SE; and (iii) MTX intake, as suggested for the latter with *Aggregatibacter* sp. by Zhang et al. in subgingival and saliva samples [15]. Moreover, the association with periodontitis was not evaluated in our cohort since data for this parameter were not collected. As the oral microbiome is unstable and affected by the diet, these results need further confirmation in an independent replicative study.

## 4. Materials and Methods

### 4.1. Subjects

Between 2014 and 2019, 125 women were recruited in a cross-sectional study from the Tatarstan prospective cohort [43] and dichotomized into two groups: one to study the different steps leading to early RA development (<1 year from diagnosis) and another one to study RA-associated factors in a replicated RA-established cohort. In non-RA individuals (*n* = 69), enriched in FDR from RA probands, CSA criteria were evaluated and CSA was established when the score was ≥3, according to the European League Against Rheumatism (EULAR) that included seven relevant parameters: joint arthralgia (<1 year), metacarpophalangeal (MCP) joint arthralgia, morning stiffness (MS) duration ≥60 min, most severe symptoms in early morning, FDR with RA, difficulty with making a fist, and positive squeeze test of MCP joints [7]. The joint symptom evaluation was performed by a rheumatologist and the evaluation was completed by magnetic resonance imaging in the case of joint symptoms (tender, swollen and MS) in the small joints of the feet and hands. RA diagnosis was settled according to the 2010 American College of Rheumatology (ACR)/EULAR criteria [44,45,46], and the DAS28-ESR was further used, with a value >3.2 implying moderate/active disease and ≤3.2 low/no disease activity [47]. Serum levels of ACPA, RF, ESR, and HLA-DRB1 SE status were collected. ACPA/RF with a titer ≥3 ULN were referred to as high [48], and SE alleles encompass DRB1*01:01, *01:02, *04:01, *04:04, *04:05, *04:08, and *10:01 [49]. The study was approved by the Ethical Committee of the Kazan State Medical Academy, Kazan, Russia (Permit nr 1/2002). Consent to conduct studies and to allow publication of the results was received from all the individuals involved in the study according to the legal requirements in Russia.

### 4.2. Microbiome Analysis

To study microbial diversity from oral mucosa, DNA was isolated from buccal swabs, amplified by PCR for 16S rRNA genes using the universal bacterial primers 27-forward (5′- AGA GTT TGA TCM TGG CTC AG -3′) and 1522-reverse (5′- AAG GAG GTG ATC CAG CCG CA -3′). The primer combination amplifies a 1500 bp 16S rDNA fragment, and extracted DNA was tested for quality (ratio 260/280: 1.5–2.0), quantity (DNA > 20–500 ng/mL, 50 μL), and MiSeq sequencing, performed in duplicates.

The raw bacterial reads were subjected to quality control using FastQC software, and further analysis of sequences was performed using QIIME v. 1.9.1 software [50]. Following quality filtering (with settings defined as follows: minimum 30 quality score over at least 75% of the sequence read; no ambiguous bases allowed; one primer mismatch was allowed), the remaining sequences are aligned with the DADA2 PacBio pipeline to generate ASVs followed by BLASTn against NCBI 16S Microbial to provide effective SILVA species-level taxonomic assignment when at least four sequences were aligned. The MicrobiomeAnalyst website (https://www.microbiomeanalyst.ca (accessed on 12 October 2022)) was used to test associations between the bacterial richness and evenness (alpha diversity), diversity (beta diversity), and species diversity (heat map, LefSE) [51].

### 4.3. Environmental Factors

Additional parameters collected included socio-demographic factors (age, childbirth, menopause, and educational level) and individual behavior factors (fish/alcohol/coffee consumption, no/active/passive tobacco smoking, and overweight/obesity status, using the World Health Organization definition with a body mass index ≥ 25 kg/m^2^ and ≥30 kg/m^2^, respectively [52]. Based on a median age of 50 years old (IQ: 48–52 years) for menopause in Russia, this threshold was used to distinguish pre- from post-menopause women [53]. Education was dichotomized into low education level (secondary and high school graduates) and high education level (university graduate).

### 4.4. Statistics

Quantitative results are expressed as the mean and interquartile (IQ) and compared using ANOVA, while categorical data were analyzed using the Fisher’s exact test. When necessary and as indicated, post-hoc corrections were applied for multiple comparisons, and the false discovery rate values (q-values) were calculated. Statistics are calculated from the MicrobiomeAnalyst website, Prism 9.4 (GraphPad Software, La Jolla, CA, USA), and the online false discovery rate calculator (https://www.sdmproject.com (accessed on 12 October 2022)). Statistical significance was assessed with two tailed *p* or q values lower than 0.05.

## 5. Conclusions

In conclusion, our results point toward a role of ACPA in controlling periodontal bacteria such as *Porphyromonas* sp. and *Aggregatibacter* sp. As a consequence, the ACPA status appeared to be important when exploring the oral microbiome, as previously suggested from the analysis of *Pg* in ACPA-positive individuals at risk of developing RA, and in RA patients [9,10]. Further analysis coupling oral microbiome analysis with ACPA could hold interesting promise regarding diagnosis and prognosis at the preclinical stage in RA-suspected individuals, together with their accompanying dental symptoms.

## Figures and Tables

**Figure 1 ijms-23-12599-f001:**
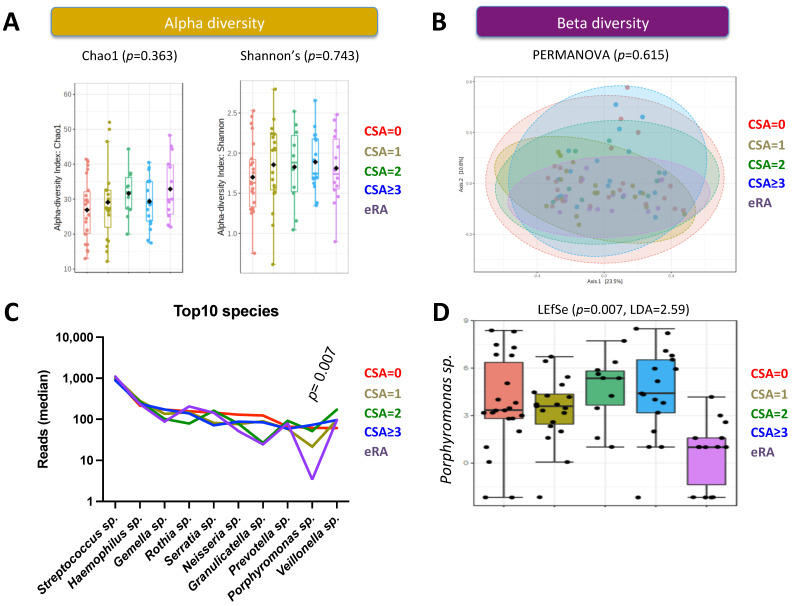
Alpha diversity, beta diversity and the oral abundance at the genus level in 15 patients with early rheumatoid arthritis (eRA), and 69 controls subdivided according to their clinically suspect arthralgia (CSA) score. (**A**) Alpha diversity as tested by Chao1 index and Shannon’s index. (**B**) Beta diversity tested by PERMANOVA (permutational multivariate analysis of variance); the percentage of variation explained by the principal component analysis (PCA) is indicated on the axes. (**C**) Top 10 genera with highest relative abundance in the 5 subgroups tested; (**D**) Box plots showing the relative abundance of the genus *Porphyromonas* sp. in controls and eRA. *p* values are based on Linear discriminant analysis (LDA) effect Size (LEfSe).

**Figure 2 ijms-23-12599-f002:**
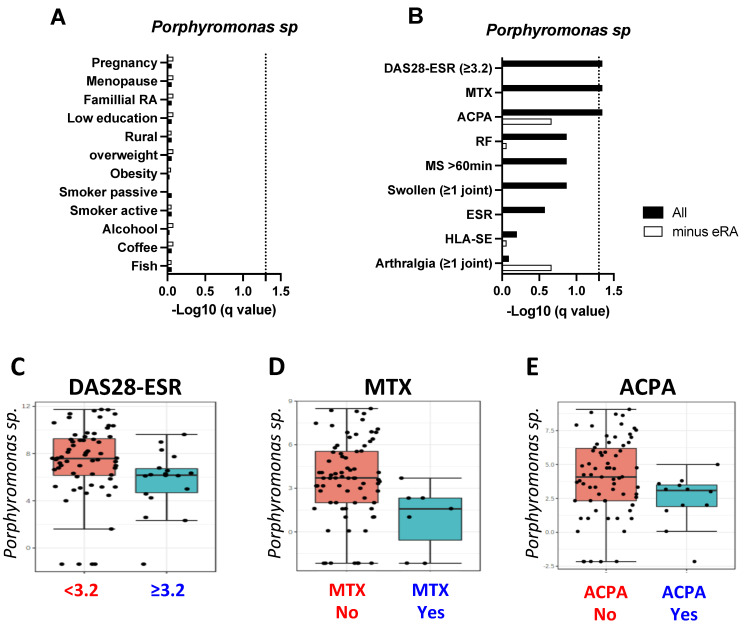
Association between oral *Porphyromonas* sp. levels and rheumatoid arthritis (RA) associated risk-factors (**A**) and pre-RA/RA clinical factors (**B**) identified by linear discriminant analysis effect size (LEfSe) analysis in 84 individuals including (black boxes) or not (white box, minus eRA) 15 patients with early-(e)RA. For graphical purposes, calculated false discovery rate q values are represented as -Log10 and the significant threshold was fixed at 1.3 (q = 0.05, dot line). Boxplots of the 3 bioclinical factors associated with *Porphyromonas* sp. level reduction within the 84 individuals: (**C**) disease activity (DAS28-ESR); (**D**) the use of methotrexate (MTX); and (**E**) anti-citrullinated peptide/protein antibodies (ACPA).

**Figure 3 ijms-23-12599-f003:**
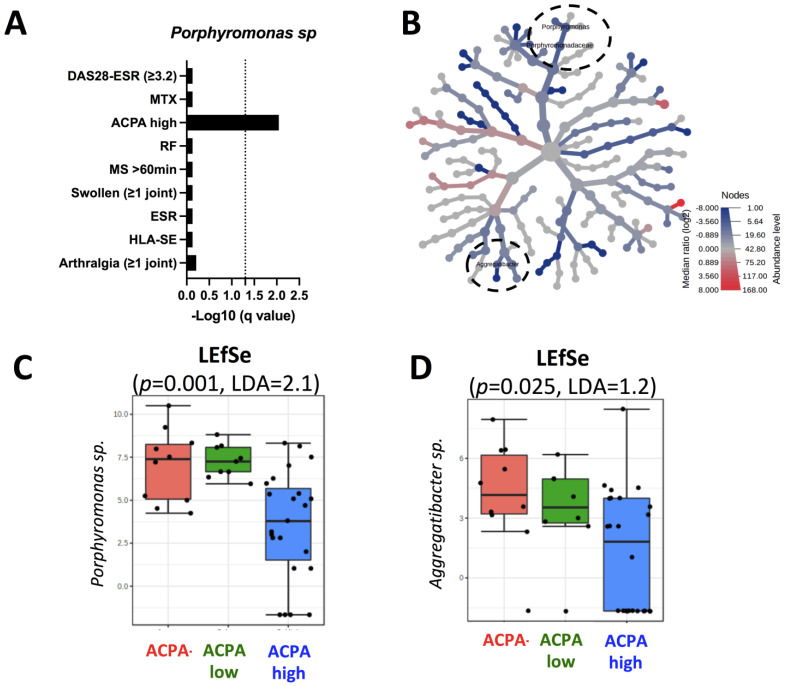
Elevated anti-citrullinated peptide/protein antibody (ACPA) levels are associated with oral *Porphyromonas* sp. and *Aggregatibacter* sp. level reduction in patients with rheumatoid arthritis (RA). (**A**) Association between oral *Porphyromonas* sp. levels and RA associated factors in 42 RA patients. (**B**) differential heat-tree matrix depicting the taxa abundance in RA patients according to the presence of ACPA at elevated levels. The red node (log 2-fold) indicates a higher abundance of the taxon, while the blue node indicates the opposite. (**C**,**D**) Boxplots showing specific differences in the relative abundances of discriminant *Porphyromonas* sp. and *Aggregatibacter* sp. according to the ACPA status: negative, low levels (upper limit of normal [ULN] = 1–2.9), and high levels (ULN > 3) (*p* < 0.05).

**Table 1 ijms-23-12599-t001:** Impact on the oral microbiome of rheumatoid arthritis (RA) and of individuals with anti-citrullinated protein/peptide antibodies (ACPA).

Technique (OTU/ASV)	Oral Location	Population	Alpha/Beta Diversity	P (Pg) and A (Aa)	Clinical Associations	References
16S (OTU)	Subgingival	54 RA, 45 PD, 44 HC	Increased richness (RA)	Similar	-	[8]
shotgun	Subgingival	48 ACPA+, 26 RA, 32 HC	Decreased richness (ACPA+)	P species (ACPA+ > RA/HC)	ACPA	[9]
16S (OTU)	Subgingival	31 eRA, 34 RA, 18 HC	Similar	P species (HC > RA)	PD (all ACPA/RF+)	[10]
16S (OTU)	Subgingival	22 RA, 19 controls	Unknown	Similar	-	[11]
16S (OTU)	Subgingival, saliva, mucosa	50 RA, 50 ACPA/RF+, 50 HC	Similar	Similar (OTU116)	-	[12]
16S (OTU)	Saliva	29 ACPA+, 27 RA, 23 HC	Decreased richness (ACPA+)	Pg (HC > ACPA+)	-	[13]
16S (ASV)	Saliva	61 eRA, 59 HC	Richness: eRA > HC	Pg and Aa (eRA > HC)		[14]
shotgun	Subgingival, saliva	54 RA, 51 HC	Unknown	Pg and A (HC > RA)	DMARDs (A species)	[15]
16S (OTU)	Saliva	110 RA, 67 OA, 155 Hc	Richness: eRA/OA > HC	Pg (RA + OA > HC)	-	[16]
16S (OTU)	Subgingival	42 RA, 47 HC	Richness: RA > HC	Aa (RA-PD > HC-PD)	PD	[17]
16S (ASV)	Mucosa	35 RA, 64 non-RA	Similar	P and A species (non-RA > RA)	-	[18]
16S (ASV)	Mucosa	15 eRA, 43 RA, 69 non-RA	Similar	P and A species (non-RA > RA)	ACPA	current study

**Abbreviations:** ASVs: amplicon sequence variants; OTU: operational taxonomic units; P: Porphyromonas species (sp.); Pg: *Porphyromonas gingivalis*; A: *Aggregatibacter* sp.; Aa: *Aggregatibacter actinomycetemcomitans*; HC: healthy controls; eRA: early-RA; RF: rheumatoid factor; OA: osteoarthritis; PD: periodontal disease.

**Table 2 ijms-23-12599-t002:** Studied population characteristics.

	Early RA	CSA = 0	CSA = 1	CSA = 2	CSA ≥ 3	*q* Values	RA
	Cohort 1						Cohort 2
Number	15	22	19	11	17	-	42
Age, median (IQ)	50 (32–58)	58 (32–63)	46 (31–56)	52 (30–62)	44 (36–59)	0.561	57 (43–78)
RA in FDR	8/15	0/22	13/19	10/11	15/17	<10^−4^	25/42
DAS28-ESR > 3.2	13/15	0/22	0/19	1/11	5/17	<10^−4^	32/42
ESR >30 mm/hr	10/15	2/22	1/19	1/11	5/17	<10^−4^	17/38
RF (>3 ULN)	11 (6)/15	0 (0)/22	1 (0)/19	1 (0)/11	4 (1)/17	<10^−4^	26 (15)/42
ACPA (>3 ULN)	10 (10)/15	0 (0)/22	1 (0)/19	1 (0)/11	1 (0)/17	<10^−4^	32 (24)/42
HLA-DRB1 SE	5/12	7/16	9/15	5/7	6/13	0.697	20/30
MS > 60 min	11/15	0/22	0/19	2/11	6/17	<10^−4^	21/42
Tender arthralgia	14/15	0/22	1/19	6/11	12/17	<10^−4^	35/42
Swollen	12/15	0/22	0/19	0/11	4/17	<10^−4^	24/42
MTX:GC:biologics:no	7:3:0:6	0:0:0:22	0:0:0:19	0:0:0:11	0:0:0:17	<10^−42^	32:19:3:5
Childbirth	13/15	18/22	15/19	10/11	13/17	0.882	35/42
Menopause (≥50 years)	9/15	16/22	8/19	7/11	5/17	0.105	27/42
Low education	8/15	12/22	9/19	4/11	6/17	0.697	23/42
Fish	9/15	19/22	14/19	4/11	12/17	0.093	31/42
Coffee	7/15	9/22	10/19	3/11	7/17	0.787	22/42
Alcohol	11/15	14/22	9/19	7/11	10/17	0.698	29/42
Smoker (active)	1/15	1/22	3/19	1/11	0/17	0.561	1/42
Smoker (passive)	2/15	8/22	9/19	4/11	3/17	0.250	15/42
Overweight	6/13	14/20	10/18	3/10	7/16	0.158	20/37
Obesity	4/13	8/20	2/18	3/10	4/16	0.533	7/37
Rural	5/15	10/22	6/18	6/11	11/17	0.093	27/42

**Abbreviations:** CSA: Clinically suspect arthralgia; RA: rheumatoid arthritis; eRA: early RA; RF: rheumatoid factor; ACPA: anti-citrullinated protein/peptide antibodies; IQ: interquartile; FDR: first degree relatives from RA probands; DAS28: joint-28 disease activity score; ESR: erythrocyte sedimentation rate; mm/hr: milliliter/hour; ULN: upper limit of normal; HAQ: health assessment questionnaire; CRP: C-reactive protein; RF: rheumatoid factor; SE: shared epitope present on HLA DRB1; MS: morning stiffness; VAS: visual analogue scale; y: years. MTX: methotrexate; GC: glucocorticoids.

## Data Availability

The data that support the findings of this study are available from the author [MIA, YR], upon reasonable request.

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
