# Peer review of "Anti-Citrullinated Peptide Antibodies Control Oral Porphyromonas and Aggregatibacter species in Patients with Rheumatoid Arthritis"

_ijms, 2022, doi:10.3390/ijms232012599_

Round 1

Reviewer 1 Report

The authors reported change of oral Porphyromonas sp and Aggregatibacter sp levels in early rheumatoid arthritis patients. However, this conclusion is questionable based on the evidences in this manuscript. First of all, oral microbiome is known to be unstable and highly affected by the dietary.  Sequencing analysis on a single cohort is too week a proof. Moreover, although the authors described the post-hoc correction for multiple comparison, there is no indication in the manuscript where they applied it. As far as I'm concerned, the P-value in Fig. 1D is unlikely to remain significant after multiple comparison correction. Therefore, rejection of manuscript is suggested. 

Author Response

First of all, the oral microbiome is known to be unstable and highly affected by the diet.  Sequencing analysis on a single cohort is too weak a proof.

Author’s response: We disagree with the reviewer's comment based on three arguments. First, two cohorts are presented in our study and the second one (confirmed-RA) can be viewed as a confirmatory cohort. Second, previous groups have reported a negative interplay between ACPA and Porphyromonas gingivalis when considering RA patients positive for ACPA that lack Porphyromonas gingivalis (Scher, A&R, 2012) or when considering ACPA positivity in RA risk groups (Cheng et al, ARD, 2021). Third, a cross-reactivity has been demonstrated between ACPA and anti-Porphyromonas antibodies (PMID: 35514991). This point was added to reinforce our observation as well as the sentence “As the oral microbiome is unstable and affected by the diet, these results need further confirmation in an independent replicative study”.

Moreover, although the authors described the post-hoc correction for multiple comparison, there is no indication in the manuscript where they applied it. As far as I'm concerned, the P-value in Fig. 1D is unlikely to remain significant after multiple comparison correction.

Author’s response: We failed to explain our statistical analysis for multiple comparisons in enough detail and apologize for that. By providing q data in Table 1, Figure 2A/B and Figure 3A, our findings become more clear, thanks for that. In addition, the material and method section was improved accordingly.

English language and style

Author’s response: We apologize for that. A native English speaker (Wesley H Brooks, University of South Florida) particularly reviewed the revised edition of the document.

Reviewer 2 Report

Dear authors, I have following questions and comments:

1.      The tittle of the manuscript “Anti-citrullinated peptide antibodies control oral Porphyromonas and Aggregatibacter species at the initiation of rheumatoid arthritis” does not correspond to the results, where the reduction of both species was observed in RA-established cohort. In early-RA patients, only the decrease in oral Porphyromonas sp was detected.

2.      In the abstract: Authors claim ACPA having a role in oral dysbiosis. Dysbiosis is defined as an imbalance of microbiota = the negative change of microbiota leading to the disease. Since Porhyromonas sp. is associated with the pathogenesis of periodontitis, disrupting the symbiosis between the oral local bacteria. I would not recommend naming the oral reduction Porhyromonas sp. as dysbiosis.

3.     I would recommend to add a short introduction about oral microbiome and periodontal pathogens.

4.     On page 2, row 55 authors write: “peptidyl arginine deiminase (PPAD)“, in row 58 authors write “PADI”, but the abbreviation PADI is not explained. Is one of the abbreviations „PPAD“ or „PADI“ a mistake?

One example is Porphyromonas gingivalis (Pg), a gram-negative anaerobe that contains a bacterial peptidyl arginine deiminase (PPAD) capable of citrullinating arginine residues in host proteins [3]. Another example is Aggregatibacter actinomycetem comitans (Aa), a facultative gram-negative and oral anaerobe, which is also able to trigger PADI expression in host neutrophils and, in turn citrullination through leukotoxin A release [4].“

Sincerely

Author Response

  1. The title of the manuscript “Anti-citrullinated peptide antibodies control oral Porphyromonasand Aggregatibacter species at the initiation of rheumatoid arthritis” does not correspond to the results, where the reduction of both species was observed in RA-established cohort. In early-RA patients, only the decrease in oral Porphyromonas sp was detected.

Author’s response: We value the reviewer’s comments and changed the title of the manuscript as follow “Anti-citrullinated peptide antibodies control oral Porphyromonas and Aggregatibacter species in patients with rheumatoid arthritis”), thanks for your suggestion to improve the title.

  1. In the abstract: Authors claim ACPA having a role in oral dysbiosis. Dysbiosis is defined as an imbalance of microbiota = the negative change of microbiota leading to the disease. Since Porhyromonas sp.is associated with the pathogenesis of periodontitis, disrupting the symbiosis between the oral local bacteria. I would not recommend naming the oral reduction Porhyromonas sp. as dysbiosis.

Author’s response: thanks for your comment and we have incorporated the reviewer’s suggestion. The term “oral microbiome changes” is used instead of “oral dysbiosis”.

  1.    I would recommend adding a short introduction about oral microbiome and periodontal pathogens.

Author’s response: we agree with your comment which was also mentioned by reviewer#3. A section related to the oral microbiome and periodontal pathogens was added in the introduction.

Introduction section: Oral microorganism colonization results from multiple parameters including diet, contact with other individuals/animals, dentition, and hygiene habits among other factors [8]. In adulthood, ecologic stability is obtained with interspecies collaborations and antagonisms in order to maintain homeostasis [9]. As reported from 200 healthy individuals and 9 oral sites, the human oral microbiome (HOMD; http://www.homd.org) is complex with over 700 prokaryotic species. The higher proportion of bacteria is reported to be from the phyla Firmicutes (e.g. Streptococcus sp, Gemella sp), while smaller proportions are observed in Bacteroides (e.g. Porphyromonas sp), Proteobacteria (e.g. Haemophilus sp, Serratia sp), and Actinobacteria (e.g. Rothia sp) [10]. Changes in the oral cavity microbiome have been associated with multiple factors including oral bacterial diseases such as periodontidis, prosthetic implant, tobacco smoking, salivary gland dysfunction, and systemic diseases [11,12].

  1.    On page 2, row 55 authors write: “peptidyl arginine deiminase (PPAD)“, in row 58 authors write “PADI”, but the abbreviation PADI is not explained. Is one of the abbreviations „PPAD“ or „PADI“ a mistake?

One example is Porphyromonas gingivalis (Pg), a gram-negative anaerobe that contains a bacterial peptidyl arginine deiminase (PPAD) capable of citrullinating arginine residues in host proteins [3]. Another example is Aggregatibacter actinomycetem comitans (Aa), a facultative gram-negative and oral anaerobe, which is also able to trigger PADI expression in host neutrophils and, in turn citrullination through leukotoxin A release [4].“

Author’s response: although similar in their effects, PPAD and PADI have distinct origins: bacterial for PPAD and mammalian for PADI. In human, 5 PADI enzymes are described. One or more are overexpressed by human granulocytes in response to a toxin produced by Aggregatibacter actinomycetemcomitans. Thanks for your advice.

English language and style

Author’s response: We apologize for that. A native English speaker (Wesley H Brooks, University of South Florida) particularly reviewed the revised edition of the document.

Reviewer 3 Report

Introduction and discussion should be improved with the suggested references

Materials and methods are well described and pertinent

Results are clearly described and very coherent with the materials and methods

CONCLUSION is  correct and interesting

However, the suggested references are for the following topics:

Authors should describe the importance of microbiologial oral microbioma

(PMID: PubMed ID 34425662)

(PMID: PubMed ID 34425659)

Author Response

Author’s response: We are grateful that the reviewer finds the manuscript to be interesting, clear, useful, and well organized. References were added as well as the description of the oral microbiome (see referee #2).

English language and style

(x) Extensive editing of English language and style required

Author’s response: We apologize for that. A native English speaker (Wesley H Brooks, University of South Florida) particularly reviewed the revised edition of the document.

Round 2

Reviewer 1 Report

Substantial revision has been made to the manuscript. Although the conclusion is not convincing enough to me, the authors has made it scientifically sound and aroused further studies in the discussions. Therefore, acceptance of the manuscript is suggested.